# Factors Affecting Intraoperative Gastro-Oesophageal Reflux in Dogs and Cats

**DOI:** 10.3390/ani12030247

**Published:** 2022-01-20

**Authors:** Ioannis Savvas, Kiriaki Pavlidou, Tilemachos Anagnostou, Eugenia Flouraki, George Kazakos, Dimitrios Raptopoulos

**Affiliations:** 1Companion Animal Clinic, School of Veterinary Medicine, Aristotle University of Thessaloniki, 546 27 Thessaloniki, Greece; kellypav@gmail.com (K.P.); tanagnos@vet.auth.gr (T.A.); gkdvm@vet.auth.gr (G.K.); drapto@vet.auth.gr (D.R.); 2Clinic of Surgery, School of Veterinary Medicine, University of Thessaly, 431 00 Karditsa, Greece; eflouraki@uth.gr

**Keywords:** anaesthesia, cat, dog, gastro-oesophageal reflux

## Abstract

**Simple Summary:**

Gastro-oesophageal reflux (GOR) is an anaesthetic complication that causes oesophageal inflammation and stricture in animals. The aim of this systematic review is to systematically identify the effect of preoperative fasting duration and drugs (anaesthetic and nonanaesthetic agents) on GOR in dogs and cats during anaesthesia. Seven studies were included in the meta-analysis. Many factors seem to affect the development of GOR in dogs and cats. However, there is a limited number of studies investigating these factors, and as the level of evidence is low-to-medium, no reliable conclusions can be extracted.

**Abstract:**

In animals, gastro-oesophageal reflux (GOR) may occur during anaesthesia, and it can lead to severe consequences such as oesophagitis and oesophageal stricture. This systematic review investigates the effect of fasting duration and anaesthetic and nonanaesthetic drugs on GOR in dogs and cats during general anaesthesia. Fifteen clinical studies met the inclusion criteria in this systematic review. In thirteen studies the population was dogs, while in two studies the population was cats. In the meta-analysis, seven studies were included. Four studies on the effect of fasting duration on GOR in dogs were included in the meta-analysis. In total, 191 dogs had a fasting duration less than 5 h, while 311 dogs had a fasting duration more than 5 h. The heterogeneity of the studies was high and statistically significant (*p* = 0.0002, I^2^ = 85%), but the overall effect was statistically nonsignificant (*p* = 0.82, odds ratio = 0.81, 95% CI 0.15, 4.26), in favour of the low fasting duration (<5 h). Concerning the effect of antacids on GOR, three studies were included in the meta-analysis. The heterogeneity of the studies was low and nonsignificant (*p* = 0.13, I^2^ = 52%) and the overall effect was statistically nonsignificant (*p* = 0.24). The low number of studies and the diverse factors affecting the incidence of reflux prevented us from reaching valuable conclusions on the risk factors for GOR.

## 1. Introduction

Gastro-oesophageal reflux (GOR) is the “silent” movement of gastric and/or duodenal contents into the oesophagus without associated eructation or vomiting, and can lead to oesophageal mucosal injury and oesophagitis in dogs and cats [1,2]. Anaesthetic agents seem to reduce the lower oesophageal sphincter (LOS) tone and this is a major factor involved in the pathogenesis of reflux [3,4,5]. The refluxate in the oesophagus can originate only from the stomach (acid reflux) or can be a mixture of gastric and duodenal contents (non-acid reflux) [6]. The incidence of GOR during anaesthesia has been investigated by the measurement of oesophageal pH. GOR occurs when the effectiveness of LOS is decreased by anaesthetic agents, gastric acidity and food withholding [3,4,5,7].

As GOR is thought to be the main cause of oesophageal inflammation and stricture in animals, it can lead to death or euthanasia [8,9]. During general anaesthesia, GOR is a common complication that occurs in 4.8% to 66.7% of dogs [6,10,11,12,13,14,15,16]. According to the literature, there are many factors that contribute to the development of GOR, such as preoperative food withholding, volume and acidity of the gastric contents, age, type of the surgical procedure, positioning of the patient and anaesthetic agents [6,10,15,17,18,19,20,21,22].

In humans, pulmonary aspiration following GOR is one of the most common causes of death related to anaesthesia. Over the years, a lot of research has been conducted to establish fasting guidelines prior to anaesthesia to reduce the risk of GOR. The traditional guideline was nil by mouth (NPO) from midnight if the surgery was scheduled for the morning, and toast with a tea (light breakfast) for the patients who were scheduled for the afternoon [23,24,25].

Thirteen years ago, the American Society of Anesthesiologists (ASA) adopted the guideline of a 6 h fast from solids and 2 h fast from clear liquids prior to an anaesthetic procedure [26,27]. Moreover, the Canadian Anaesthetists’ Society recommends a total fast of no less than five hours and suggests that policies be constructed within individual departments [28].

In human medical literature, there are two large systematic reviews with meta-analyses in the Cochrane Collaboration on preoperative fasting. According to the first one, there is no evidence that a shortened fluid fast can increase the risk of aspiration, regurgitation, or mortality compared to the standard NPO fasting practice in adults [29]. In the second one on preoperative fasting in children, no evidence was found that children who do not receive any fluids for more than 6 h preoperatively benefit compared to the children who had free access to fluids for up to 2 h preoperatively [30].

Another factor that seems to affect the development of GOR in humans is the volume and acidity of gastric content (GC), and several reports have investigated a potential correlation between them [31,32,33,34,35,36,37,38]. According to Hardy et al. [37], there is no correlation between the GC volume and the incidence of reflux at induction of anaesthesia. Moreover, it has been reported that a prolonged duration of fasting does not guarantee a decrease in the volume of GC, and a light breakfast 2–3 h before surgery does not change the pH and the volume of GC [31].

In dogs, although there are no clearly proposed fasting guidelines before induction of anaesthesia, there is evidence that an increased duration of preoperative fasting may cause a high incidence of GOR [10,15,18]. Other suggested guidelines are withholding of food and water for approximately 12 h before surgery [10,12,13,21,39,40,41,42] or overnight [6,22,43]. Furthermore, the effect of various types of food given on two different preoperative fasting times, 3 or 10 h beforehand, on GC volume and acidity has been studied in dogs. This study suggests that feeding dogs with canned food at a half daily rate 3 h before anaesthesia nonsignificantly increased the GC volume, while the GC acidity decreased significantly [18].

On the other hand, in a recent study by Viskjer et al. [14], a 3 h fasting period was associated with significantly greater odds for reflux compared to an 18 h fasting period. However, in the same study, the pH of the refluxate was significantly higher in the 3 h fasting group. Another study in which the dogs were fed with canned food at half daily rate 12 h before the induction of anaesthesia reported an overall incidence (44.1%) of GOR in dogs [16].

A factor that seems to affect the incidence of GOR in dogs is the different types of drugs (anaesthetic/nonanaesthetic agents). Studies have investigated the effect of opioids such as morphine [6] and pethidine [21], injectable anaesthetic agents, (propofol and thiopentone) [42], inhalant anaesthetic agents, (halothane, isoflurane and sevoflurane) [22] on the incidence of GOR. Except for these agents, other drugs have also been investigated for their impact on GOR in dogs, such as D2 receptor antagonists [13,43], antiemetics [41] and antacids [12,39,40].

In anaesthetized cats, there is little evidence about the effect of different drugs on the LOS and on the incidence of GOR. There are studies investigating the effect of acepromazine, atropine, pethidine [4], propofol, thiopentone, alphaxalone, ketamine and xylazine [3] on LOS, and other studies about the action of induction agents [20] and antacids [44] on GOR.

Since there are no specific guidelines for reducing the incidence of GOR during anaesthesia in veterinary medicine, the main aim of this systematic review with meta-analysis is to systematically identify the effect of preoperative fasting duration and of drugs (anaesthetic and nonanaesthetic agents) on GOR in dogs and cats during anaesthesia.

## 2. Methods

The Systematic Review Protocol For Animal Intervention Studies (SYRCLE) [45] was used for the study protocol.

### 2.1. Type of Studies

We included controlled studies, randomized or not, experimental or with client-owned animals, which systematically identify the effect of the preoperative fasting and drugs (anaesthetics, opioids, antacids, D2 receptor antagonists and antiemetics) on the incidence of GOR in anaesthetised dogs and cats. Reviews and non-English-language publications were not included.

### 2.2. Population/Species Studied

The target species were dogs and cats, of all ages.

### 2.3. Interventions

In this review, seven interventions were evaluated:The duration of preoperative fasting (control more than 5 h, intervention less than 5 h).The induction agents (thiopentone vs. propofol).The inhalant agents for maintenance of anaesthesia (halothane vs. isoflurane or sevoflurane).The use of opioids (control no, intervention yes).The use of antacids (control no, intervention yes).The use of D2 receptor antagonists (control no, intervention yes)The use of antiemetics (control no, intervention yes).

### 2.4. Outcome Measures

The incidence of GOR.

### 2.5. Search Method

Four electronic databases were searched:MEDLINE via PubMed.Web of Science.CAB Abstracts.SCOPUS.

CAB Abstracts was searched through Web of Science.

The search string was:(anesthe* OR anaesthe*) AND (reflux OR "gastric content" OR "gastric volume") AND (dog OR cat) AND (GOR OR GER)(1)

All publications were searched until 15 January 2022.

### 2.6. Selection of Studies

Two groups with two authors each (IS-KP and TA-EF) evaluated the results of the search output. The two groups resolved any discrepancies with collaboration and critical discussion. In the first selection phase, the title and abstract of the studies were evaluated, and then the selected studies were critically read in full (second phase). Any authors of this review were excluded from the evaluation of any eligible study in which they were also authors.

### 2.7. Data Extraction and Management

The two groups of reviewers independently extracted the details of the eligible studies. Data extracted were:Authors, title, year of publication, journal.Number of animals in intervention and control groups.Dogs/cats, age, weight, status ASA.Outcome measures.Presence of any other outcome measures.Excluded animals (dropouts).

### 2.8. Assessment of Risk of Bias in Included Studies

The SYRCLE’s Risk of Bias tool [46] was used for the assessment of the included studies. The following details were agreed on:In randomised studies, if the randomisation method was not mentioned, the risk of bias was set to unclear.Random housing of the experimental animals, the animal assessors’, and animal selection blindness were judged as low or unclear risk of bias, because we assumed that this was mostly irrelevant to our review.

### 2.9. Data Analysis

A dedicated software (Review Manager/RevMan Version 5.3. Copenhagen: The Nordic Cochrane Centre, The Cochrane Collaboration, 2014) was used for the introduction, storage, analysis, and synthesis of data to produce the meta-analysis. The outcome was dichotomous and was analysed with the Mantel–Haenszel method, with a random effects model. Subgroup analysis was performed in one case; in the duration of the fasting time outcome, two subgroups were analysed regarding the use of opioids in the anaesthetic protocol. Effect measures are presented as odds ratios. Heterogeneity and overall effects were calculated. Statistical significance was set to α = 0.05.

## 3. Results

A total of 474 papers were retrieved. PubMed search returned 116 results, Scopus 193 and Web of Science/CAB Abstracts 165. After removing duplicates, 291 papers remained. The first selection phase revealed 44 papers eligible for further evaluation. The second selection phase revealed 15 papers, which were included in this review, and 29 papers were excluded. From the 15 included studies, 7 studies were evaluated for meta-analysis (Figure 1).

### 3.1. Description of the Included Studies

Fifteen clinical studies met the inclusion criteria in this systematic review. In 13 studies the population was dogs, while in two studies the population was cats. In all studies the main outcome was the incidence of GOR. In most of the clinical studies, the animals were submitted to surgical procedures, mainly orthopaedic. In cats, the animals were submitted to castration, skin tumour excision or orthopaedic procedures in the first study [20] and to dental procedures in the second one [44].

Overall, 4 out of 13 studies on dogs investigated the effect of preanaesthetic fasting on GOR (Table 1) [10,14,15,47], 1/13 studies was about the effect of induction agents on GOR (Table 2) [42], 1/13 studies was about the effect of inhalant agents for the maintenance of anaesthesia on GOR [22] (Table 3), 1/13 was about the effect of morphine on GOR [6] (Table 4), 3/13 were about the effect of antacids on GOR [12,39,40] (Table 5), 2/13 were about the effect of metoclopramide on GOR [13,43] (Table 6), and 1/13 was about the effect of maropitant on GOR [41] (Table 7). One out of two studies in cats investigated the effect of omeprazole on GOR [44], and the other one investigated the effect of induction agents on GOR [20] (Table 8).

### 3.2. Risk of Bias of the Included Studies

The risk of bias was judged to be unclear or low in most of the studies. The risk-of-bias tables are shown in Figure 2 and Figure 3.

### 3.3. Characteristics of the Excluded Studies

Twenty-nine studies were excluded after critically evaluating them (26 in dogs and 3 in cats):One was a retrospective study of the risk factors and prevalence of regurgitation in dogs [1].Five studies, three in dogs [5,7,48] and two in cats [3,4], were about the effect of anaesthetic drugs on LOS pressure.One was a retrospective study about the oesophageal strictures in dogs and cats [8],another study was about the effect of the body position (dorsal or lateral) on the incidence of GOR in dogs [17]One experimental, cross-over study was about the effect of food withholding duration on gastric content and pH without any incidence of GOR [18].One study was about the effect of opioids on GOR, there was no control group [21].One study was in Portuguese language [49].One study was a case report about the incidence of rhinitis after intraoperative GOR in a dog [50].Two studies were review articles about GOR in dogs [51,52].One clinical study was about the incidence of GOR in kittens, comparing two airway devices [53].Three studies were about the incidence of postoperative regurgitation in dogs [2,54,55].One clinical study was about the effect of maropitant on preventing postoperative GOR in dogs [56].Two clinical studies investigated the correlation between the ovarian cycle and the incidence of GOR in dogs [11,57].One clinical study was about the correlation between the shape of the dog’s chest and GOR [58].One study was a prospective, observational study for intraoperative GOR but with no interventions [59].One study was about passive regurgitation in dogs [60].One was a randomized, controlled clinical study but there was no control group, without opioids [61].One study investigated the incidence of GOR in comparison with the type of surgical procedure [62].One study was about the incidence of GOR in brachycephalic dogs in comparison with nonbrachycephalic dogs [63].One clinical study described a noninvasive model of gastro-oesophageal reflux in dogs [64].One was a review article about oesophagitis and stricture formation in dogs and cats [65].

### 3.4. The Effect of Fasting Duration on GOR

Four studies on the effect of fasting duration on GOR in dogs were included in the meta-analysis (Figure 4). Data from a total of 502 animals were analysed. Overall, 191 dogs had a fasting duration less than 5 h, while 311 dogs had a fasting duration more than 5 h. In the publication by Galatos et al. [10], the dogs had been allocated into eight groups, one with a <5 h fasting duration, and seven with a >5 h fasting duration. Moreover, in one group, pethidine was administered as premedication, so this group was removed, in order to perform sensitivity analysis. The remaining six groups of >5 h fasting were merged into one group, in order for this paper to be included in the meta-analysis. In the publication by Tsompanidou et al. [47], the dogs had been allocated into three groups, two with a <5 h fasting and one with >5 h fasting. In order to include this paper in the meta-analysis, the two <5 h groups were merged into one. The heterogeneity of the four studies was high and statistically significant (I^2^ = 85%, tau^2^ = 2.28, chi^2^ = 19.44, df = 3, *p* = 0.0002,), and the overall effect was statistically nonsignificant (z = 0.25, *p* = 0.80, odds ratio = 0.81, 95% CI 0.15, 4.26), in favour of the low fasting duration (<5 h).

Because of the high heterogeneity, the four publications were rigorously inspected. It was obvious that two of them (Viskjer et al. [14] and Tsompanidou et al. [47]) were in favour of >5 h fasting, whereas the other two (Galatos et al. [10] and Savvas et al. [15]) were in favour of <5 h fasting. The main difference between these two groups of publications was the use of opioids in the anaesthetic protocol. Thus, a sensitivity analysis (subgroup analysis) was performed, which revealed a statistically significant difference between the subgroups (chi^2^ = 14.11, df = 1, *p* = 0.0002, I^2^ = 92.9%). When opioids were used (Viskjer et al. [14] and Tsompanidou et al. [47]), the overall effect was statistically significant in favour of >5 h fasting (z = 2.40, *p* = 0.02, odds ratio = 2.98, 95% CI 1.22, 7.30), with low heterogeneity (tau^2^ = 0.17, chi^2^ = 1.67, df = 1, *p* = 0.20, I^2^ = 40%), whereas when opioids were not used (Galatos et al. [10] and Savvas et al. [15]) the overall effect was statistically significant in favour of <5 hours fasting (z = 2.9, *p* = 0.004, odds ratio = 0.17, 95 CI 0.05, 0.56), again with very low heterogeneity (tau^2^ = 0.00, chi^2^ = 0.65, df = 1, *p* = 0.42, I^2^ = 0%).

### 3.5. The Effect of Induction Agent on GOR

One study was included in this systematic review on the effect of induction agents on GOR in dogs and one in cats. Raptopoulos et al. [42] investigated the effect of two injectable anaesthetic agents on the incidence of GOR in dogs undergoing surgery. Overall, 6 out of 34 dogs in the thiopentone group and 17 out of 34 dogs in the propofol group exhibited GOR. The difference in proportions between the two groups was statistically significant (*p* < 0.02). The effect of these two injectable anaesthetic agents had also been studied in cats [20]. In a population of 50 cats, 4/25 in the thiopentone group and 3/25 in the propofol group experienced GOR with statistically nonsignificant difference of proportions detected between the groups.

### 3.6. The Effect of the Inhalant Anaesthetic Agents for the Maintenance of Anaesthesia on GOR

The influence of three different inhalant anaesthetics on GOR in dogs during anaesthesia was investigated in the clinical study of Wilson et al. [22]. Ninety dogs were randomly allocated in the three groups (halothane, isoflurane and sevoflurane group). According to the results of the study, GOR was detected in 14/30 in the isoflurane group, 19/30 in the halothane group and 18/30 in the sevoflurane group. However, the three groups differed nonsignificantly with a *p* = 0.39.

### 3.7. The Effect of Opioids on GOR

Only one clinical study on the effect of opioids on the incidence of GOR met the inclusion criteria of this systematic review. Wilson et al. [6] studied whether the administration of morphine prior to anaesthesia in dogs undergoing surgery had an impact on the occurrence of GOR. The proportions of dogs with GOR were 8/30 (27%), 15/30 (50%), and 18/30 (60%) for the three groups (morphine dosages of 0, 0.22, and 1.1 mg/kg) respectively. A statistically significant difference (*p* = 0.03) was detected between the control group (morphine at 0 mg/kg) and the intervention groups (administration of morphine).

### 3.8. The Effect of Antacids on GOR

Omeprazole and esomeprazole have been studied concerning their effect on the incidence of GOR. Three studies in dogs and one in cats were included in this review, with the studies in dogs included in the meta-analysis (Figure 5). Data from a total of 154 animals were analysed. In total, 88 dogs received antacids, while 66 did not receive them. Furthermore, 30 dogs out of 88 had an episode of GOR and 32 out of 66 did not. The heterogeneity of the studies was low and nonsignificant (*p* = 0.13, I^2^ = 52%) and the overall effect was statistically nonsignificant (*p* = 0.24).

In cats, the administration of omeprazole preanaesthetically did not significantly affect the incidence of GOR in a relevant study (*p* = 0.057) [44]. In addition, 9 of 27 cats (33.3%) had at least one episode of GOR during anaesthesia, 7 out of 14 cats (50.0%) in the placebo group, and 2 out of 13 cats (15.4%) in the omeprazole group.

### 3.9. The Effect of D2 Receptor Antagonists on GOR

Among the included studies in this systematic review, Wilson et al. [6] investigated the effect of metoclopramide on the incidence of GOR in 52 dogs undergoing elective surgery. Twelve out of 18 (67%) dogs in the saline group, 7 out of 16 (44%) in the low-dose metoclopramide group and 6 out of 18 (33%) in the high-dose metoclopramide group had an episode of GOR. The control group differed significantly from the high-dose metoclopramide group (*p* = 0.045). Only 7 dogs out of 90 (7.8%) developed reflux episodes in the study of Favarato et al. [13]: 4/30 (13.3%) in the control group, 2/30 (6.66%) in the ranitidine group, and 1/30 (3.33%) in the metoclopramide group. Nonsignificant differences were found in the number of dogs that presented GOR episodes among the groups.

### 3.10. The Effect of Antiemetics on GOR

Concerning the effect of maropitant on GOR, 26 dogs were recruited in a clinical study [41]. In total, 6 out of 13 (46%) dogs in the control group and 4/13 (31%) in the maropitant group experienced GOR during anaesthesia. The difference was statistically nonsignificant (*p* = 0.68).

## 4. Discussion

In this review, we aimed to systematically identify, appraise, and synthesize reliable evidence from valid research about factors that affect the incidence of GOR in anaesthetised dogs and cats. According to our literature search, this is the first systematic review with meta-analysis to evaluate the effect of specific factors on the incidence of GOR in these species.

As there is a variety of factors that can influence the development of GOR, we decided to include in this systematic review seven interventions (fasting duration, induction agents, inhalant agents, opioids, D2 receptor antagonists, antacids and antiemetics). We defined the control and the intervention group for each of the seven interventions as follows: for the incidence of fasting duration, the control group was set as a fasting duration less than 5 h and the intervention group as a fasting duration more than 5 h. According to the literature, there are studies in which the animals had a food withholding for more than 6–7 h (usually 12 h or overnight) and studies with the fasting duration 2–4 h. So, we decided to use 5 h as a cut-off point for the fasting duration. In the studies investigating the induction and inhalant agents, we used thiopentone and halothane as control groups, because these drugs are older. For all the other interventions concerning drugs, the control group was set as the group without administration of the intervention drug, and the intervention group was set as the group with administration of the intervention drug. In this review, the results of 15 studies were evaluated and 7 studies were included in the meta-analysis.

We used the SYRCLE’s risk of bias tool for animal studies for the appraisal of the studies. This tool is in agreement with the Cochrane RoB tool, but it is important to mention that this item is quite difficult to assess in animal intervention studies, because protocols for animal studies are not yet registered in a central, publicly accessible database. Although this tool is not totally validated, its use can facilitate and improve the critical appraisal of animal studies [46].

Many interventions which affect the incidence of GOR were found in other studies, but as they did not meet the inclusion criteria of this systematic review, these studies were excluded. Concerning the 29 studies that were excluded, two points should be discussed. Firstly, several studies that investigated the influence of different anaesthetic drugs (xylazine, acepromazine, atropine, meperidine, thiopentone, propofol, alfaxalone, ketamine, halothane, isoflurane) on the LOS and barrier pressure in dogs and cats, but not on the incidence of GOR, were excluded from the present review [3,4,5,7,17,48]. Although these studies were excluded, it is important to mention that the factors affecting LOS function may be risk factors for the development of GOR. Anagnostou et al. [58] have published three very interesting clinical trials on GOR. In the first one, the effect of the different shape of the chest (large-sized, deep-chested versus small-sized, barrel-chested) on the incidence of GOR was studied. According to this study, the incidence of GOR during surgery in sternal recumbency is statistically significantly higher in large-sized, deep-chested dogs than in small-sized, barrel-chested dogs. As the effect of the size and type of the chest on GOR was investigated only in this study, we decided to exclude it from the present systematic review. Moreover, there are other two studies by Anagnostou et al. on the effect of hormones, endogenous progesterone and oestradiol 17β [11], and the stage of the ovarian cycle and pregnancy on GOR [57]. Concerning the first study, it was not likely that the above hormones affect the incidence of GOR during the ovarian cycle. According to the second one, the occurrence of GOR was higher in pregnant dogs. As these studies did not meet our inclusion criteria, we excluded them.

The fasting duration before anaesthesia seems to be an important factor that influences the incidence of GOR, and there is a lot of discussion about the proper fasting guidelines in humans and animals. According to our meta-analysis, two out of four studies are in favour of a fasting duration less than 5 h to prevent the incidence of GOR in dogs (more than 5 times reduction in the risk of GOR), which is in agreement with the evidence from medical literature [29,30,66]. Fasting guidelines in humans show that a 2 h fast from fluids and 6 h from solids is safe for healthy adults [67]. In the study by Galatos et al. [10], an increased duration of preoperative fasting was associated with a high incidence of reflux. The clinical study by Savvas et al. [15] is also in favour of fasting duration less than 5 h. These findings are in agreement with the GOR incidence discussed in another study, investigating the effect of fasting on GC volume and pH, in which the administration of canned food at half daily rate 3 h before anaesthesia increased the risk of GOR is dogs, but this would be an effect of chance alone (nonsignificant effect) [18].

In contrast, the two other studies (Viskjer et al. [14] and Tsompanidou et al. [47]) of this meta-analysis revealed a totally different outcome. A light meal 3 h before anaesthesia was associated with a significantly higher incidence of GOR in comparison with overnight food withholding. The authors of the first study (Viskjer et al. [14]) gave some explanations as to why their results differed from other studies. The large volume of the meal and the different composition of food (the amount of proteins and fibres were higher in comparison with the food in the study by Savvas et al. [18]) may have been responsible for the different findings in comparison with the other two studies. Moreover, the administration of an opioid as preanaesthetic medication in combination with the amount and type of food may influence gastric motor function. Another important point in the design of that study is that the dogs that underwent pelvic limb surgery received an additional dose of morphine epidurally prior to surgery. All these limitations could explain the different findings in that study regarding the appropriate duration of fasting. Likewise, in the other study (Tsompanidou et al. [47]) pethidine was used to premedicate the dogs. From the subgroup analysis, it seems that the use of opioids in the anaesthetic protocol may lead to a threefold increase in the risk of GOR. It has been shown that opioids can induce GOR [6,21], so clinicians must always bear in mind this effect when they use opioids.

The second meta-analysis concerned the effect of antacids on the risk of GOR in dogs. The common point in the study design of those three studies was that there was a control group (no antacid) and two interventions groups (different doses of the same drug). In the meta-analysis, we arbitrarily combined the two interventions groups into one. Although the heterogeneity of the studies was low and nonsignificant, the overall effect was also statistically nonsignificant and therefore there is not enough evidence that the administration of antacids may decrease the risk of GOR in dogs during anaesthesia.

According to our literature search, not only antacids but also antiemetics, such as maropitant, and D2 receptor antagonists, such as metoclopramide, influence the incidence of GOR. The findings on the effect of metoclopramide on GOR during anaesthesia are different between the two relevant studies by Favarato et al. [13] and Wilson et al. [43]. The lack of a beneficial effect of metoclopramide alone or with ranitidine in preventing GOR in the study by Favarato et al. [13] may be due to the anaesthetic protocol used, which was not anticipated to contribute to a high occurrence of GOR, as an opioid was not included. Their findings are in contrast to the results of the study by Wilson et al. [43], which showed a reduced risk of GOR after the administration of metoclopramide, however morphine was part of the anaesthetic protocol as preanaesthetic medication in the study by Wilson et al. [43].

Different induction agents, thiopentone and propofol [42], or different inhalant agents for the maintenance of anaesthesia halothane, isoflurane and sevoflurane [22], seem to affect GOR differently, but a definite conclusion cannot be drawn, because only two studies were included. However, concerning the effect of injectable anaesthetics, thiopentone is not so widely used anymore in clinical practice, and therefore it should be kept in mind that when propofol is used, the risk of GOR is increased.

Regarding the opioids, contradictory results have been reported by different studies. There is little evidence on the effect of opioids on GOR. Only two studies have been published on the effect of morphine [6] and pethidine [21] on GOR in dogs. Morphine increases the incidence of GOR during anaesthesia, while the use of pethidine decreases the incidence of GOR when compared to morphine premedication [21]. In the study on morphine, a prolonged period of food withholding (10–30 h) was implemented, and this may have contributed to the high incidence of GOR [6]. According to the authors, the food was withheld from the dogs for a mean of 18 h prior to anaesthesia and this is in agreement with other studies [10,19]. However, no animal that had a fasting duration between 2–4 h, had an episode of GOR. So, it seems that the association between fasting duration and the effect of morphine on GOR is a subject that requires further investigation.

The present review has some limitations. The major limitation is the small number of the included studies. Despite the fact that the overall number of the studies on GOR, regurgitation and LOS in dogs is not small, there is high variability among the studies. They differ on the drugs that are used, on the technique for the assessment of GOR [68], and also on the factors studied that can influence the occurrence of GOR. As a result, the limited number of the included studies leads to a limited number of studies for a meta-analysis. Moreover, the number of studies about the incidence of GOR in cats is extremely low. It is obvious that further research is needed in the field of gastro-oesophageal reflux in companion animals during anaesthesia. Attempts should be made to establish specific guidelines on preoperative fasting, like in human medicine, that should be followed by the anaesthetists in order to decrease the incidence of GOR during anaesthesia in dogs and cats.

## 5. Conclusions

This review reveals that:There are many factors affecting the development of GOR during anaesthesia in dogs.There is a limited number of studies investigating each one of these factors.The evidence is low-to-medium and cannot allow for the extraction of reliable conclusions on how these factors affect the development of GOR during anaesthesia.Evidence in cats is even more scarce.

More studies are necessary for safer conclusions.

## Figures and Tables

**Figure 1 animals-12-00247-f001:**
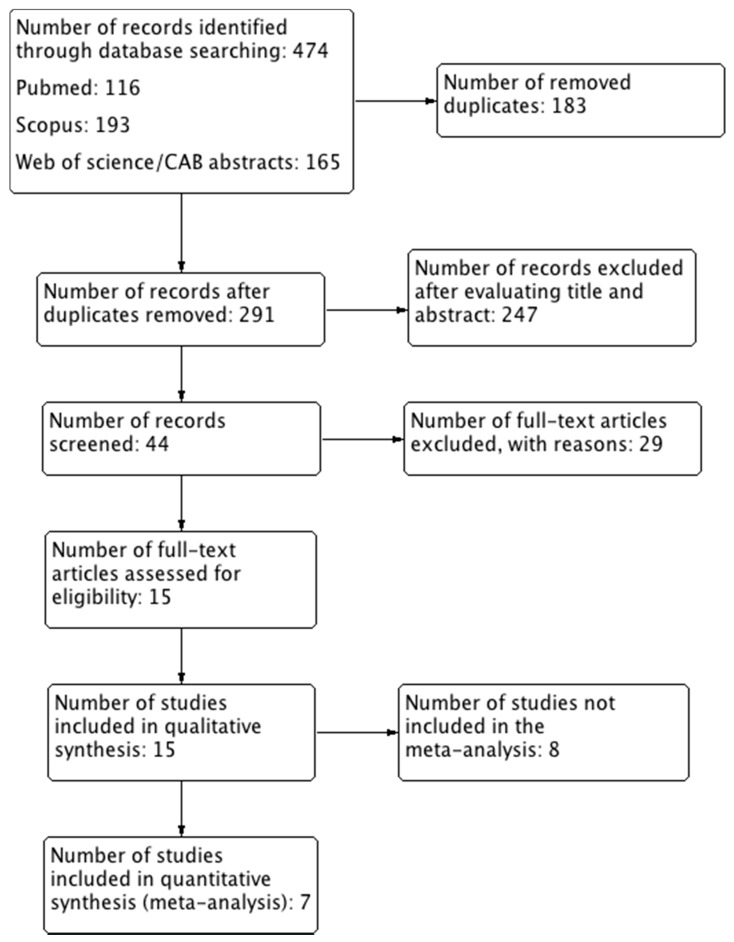
Study flow diagram.

**Figure 2 animals-12-00247-f002:**
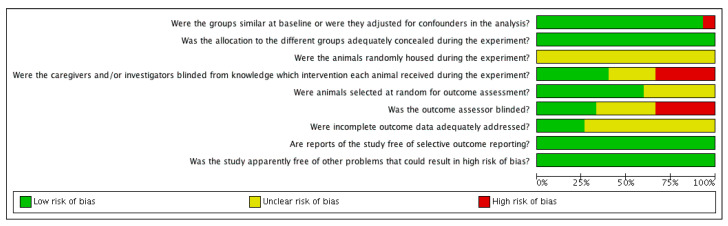
Methodological quality graph: review authors’ judgements about each methodological quality item presented as percentages across all included studies.

**Figure 3 animals-12-00247-f003:**
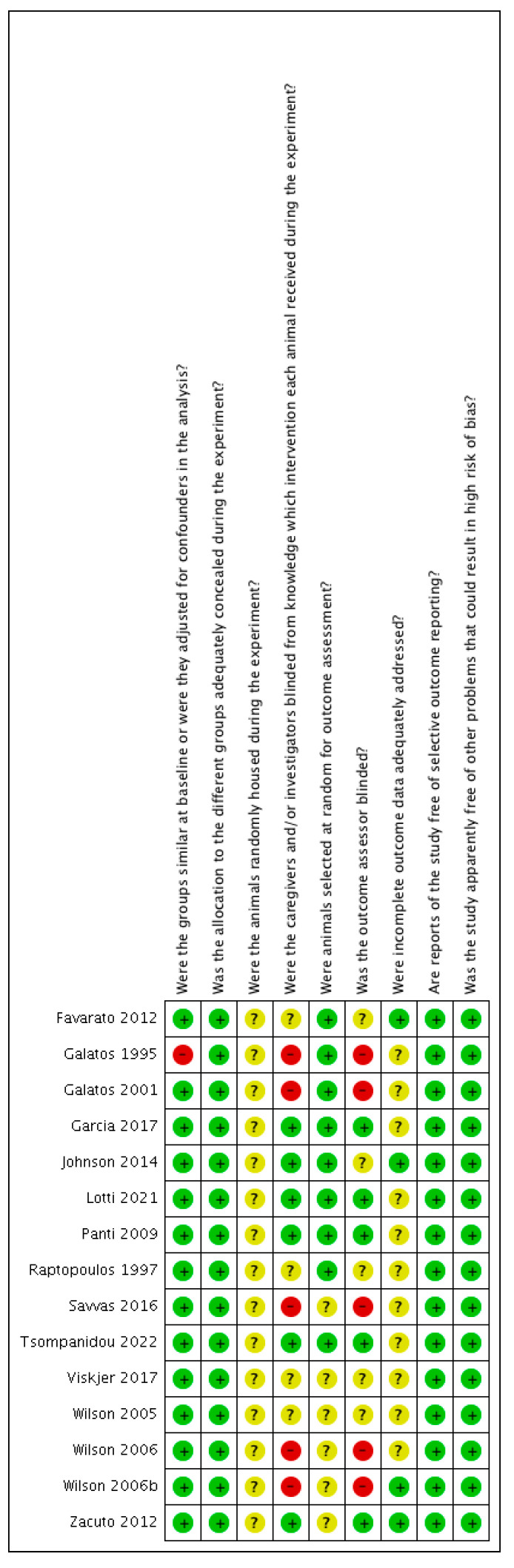
Methodological quality summary: review authors’ judgements about each methodological quality item for each included study. ?: unclear risk, −: high risk; +: low risk.

**Figure 4 animals-12-00247-f004:**
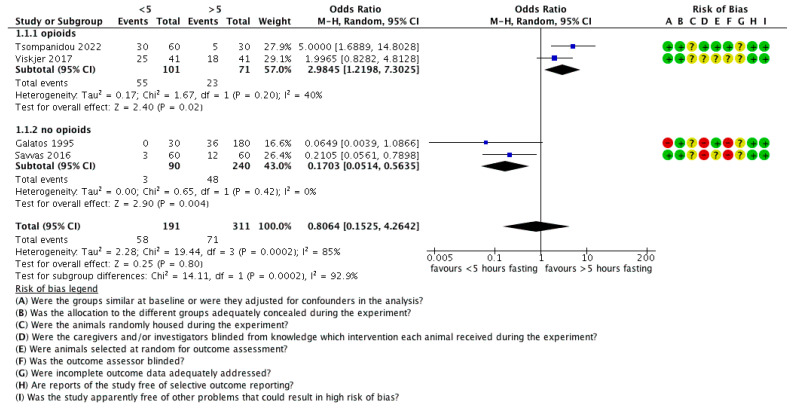
Forest plot of the effect of fasting time on GOR. CI: confidence interval. ?: unclear risk, −: high risk; +: low risk.

**Figure 5 animals-12-00247-f005:**
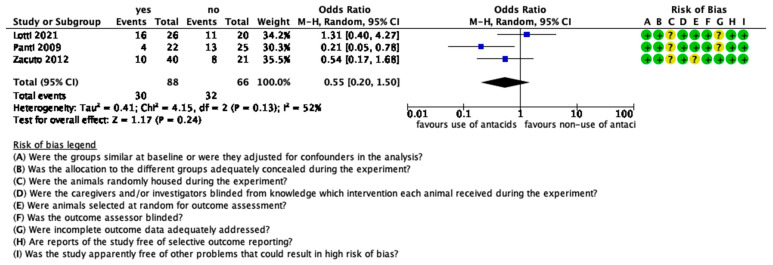
Forest plot of the effect of antacids on GOR. CI: confidence interval. ?: unclear risk; +: low risk.

**Table 1 animals-12-00247-t001:** Intervention: duration of fasting. Characteristics of the studies in dogs.

Reference	Type of Study/Design	Animals (Dogs)	Control/ Intervention	Type of Drugs Used	Procedure
Galatos et al., 1995 [10]	prospective cohort clinical study	240 (125 females, 115 males), age 6 months–9 years, weight 2.5 to 46 kg	2–4 h/12–18 h	propionylpromazine, atropine, xylazine, pethidine, diazepam, thiopentone, halothane	nonabdominal, nonthoracic
Savvas et al., 2016 [15]	randomized clinical trial	120, age 1–8 years, weight 4.6–42 kg	3 h/10 h	propionyl promazine, thiopentone, halothane	nonabdominal, nonthoracic, no head tilt
Viskjer 2017 et al. [14]	prospective, randomized, controlled clinical trial	82, age 48.39 ± 35.31 months, weight 27.37 ± 13.14 kg	3 h/18 h	acepromazine, methadone, morphine, propofol, isoflurane, morphine epidurally in some dogs	orthopaedic surgery
Tsompanidou et al., 2022 [47]	prospective, randomized, controlled clinical trial	90 (37 females, 53 males), age 1–10 years, weight 5–39 kg	3 h/12 h	acepromazine, pethidine, propofol, isoflurane	nonabdominal, nonthoracic

**Table 2 animals-12-00247-t002:** Intervention: induction agent. Characteristics of the study in dogs.

Reference	Type of Study/Design	Animals (Dogs)	Control/Intervention	Fasting Duration	Procedure
Raptopoulos et al., 1997 [42]	randomized clinical trial	68 (35 females, 33 males), age 6 months–9 years, weight 2.5–50 kg	thiopentone/propofol (maintenance with halothane)	12–18 h	soft tissue, orthopaedic, imaging

**Table 3 animals-12-00247-t003:** Intervention: inhalant agents. Characteristics of the study in dogs.

Reference	Type of Study/Design	Animals (Dogs)	Control/Intervention	Fasting Duration	Procedure
Wilson et al., 2006 [22]	randomized, prospective clinical trial	90 dogs, various ages and weights	halothane/isoflurane or sevoflurane	overnight	orthopaedic surgery

**Table 4 animals-12-00247-t004:** Intervention: use of opioids. Characteristics of the study in dogs.

Reference	Type of Study/Design	Animals (Dogs)	Control/Intervention	Fasting Duration	Procedure
Wilson et al., 2005 [6]	randomized, prospective clinical trial	90, age 4.8 ± 2.4 years, weight 32 ± 2.7 kg	no morphine/morphine at 0.22 mg/kg or 1.10 mg/kg	overnight	orthopaedic surgery

**Table 5 animals-12-00247-t005:** Intervention: use of antacids. Characteristics of the studies in dogs.

Reference	Type of Study/Design	Animals (Dogs)	Control/Intervention	Fasting Duration	Procedure
Panti et al., 2009 [12]	randomized, blinded, controlled clinical trial	47, weight 32.7 ± 14.3 kg	no omeprazole/omeprazole 1 mg/kg	>12 h	pelvic limb surgery
Zacuto et al., 2012 [40]	prospective, randomized, blinded, placebo-controlled study	61, age 4.9 ± 3.4 years, weight 25.0 ± 12.9 kg	saline/esomeprazole 1 mg/kg or esomeprazole 1 mg/kg and cisapride 1 mg/kg	>12 h	orthopaedic surgery
Lotti et al., 2021 [39]	prospective, randomised, blinded, clinical trial	55, age 5–60 months, weight 21 ± 6.9 kg	no omeprazole/omeprazole 1 mg/kg or omeprazole two doses of 1 mg/kg each	>12 h	ovariectomy

**Table 6 animals-12-00247-t006:** Intervention: use of metoclopramide. Characteristics of the studies in dogs.

Reference	Type of Study/Design	Animals (Dogs)	Control/Intervention	Fasting Duration	Procedure
Wilson et al., 2006 [43]	prospective, randomised, clinical trial	52, ages and weights not mentioned	saline/metoclopramide 0.4 mg/kg or 1.0 mg/kg	overnight	orthopaedic surgery
Favarato et al., 2012 [13]	randomized, controlled clinical study	90 (female), age 0.5–9 years, weight 1.5–34 kg	no metoclopramide/metoclopramide 1 mg/kg or ranitidine 2 mg/kg	12 h	ovariohysterectomy

**Table 7 animals-12-00247-t007:** Intervention: use of maropitant. Characteristics of the study in dogs.

Reference	Type of Study/Design	Animals (Dogs)	Control/Intervention	Fasting Duration	Procedure
Johnson et al., 2014 [41]	randomized, blinded, prospective clinical study	26 (18 females, 8 males), age 3.1 ± 3.1 years, weight 20.5 ± 11.4 kg	saline/maropitant 1.0 mg/kg	>12 h	soft tissue and orthopaedic surgery

**Table 8 animals-12-00247-t008:** Characteristics of the two studies in cats.

Reference	Type of Study/Design	Animals (Cats)	Control/Intervention	Fasting Duration	Procedure
Galatos et al., 2001 [20]	randomized, clinical trial	50 (2 females, 48 males), age 6 months–8 years, weight 2.9–6.3 kg	thiopentone/propofol	overnight	castration, skin tumour excision, orthopaedic surgery
Garcia et al., 2017 [44]	prospective, blinded, placebo-controlled, randomized clinical trial	27, age 8.6 ± 3.8 years, weight 5.5 ± 0.8 kg	placebo/two different doses of omeprazole	12 h	dental procedures

## Data Availability

Data are contained within the article.

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
