# Peer review of "Factors Affecting Intraoperative Gastro-Oesophageal Reflux in Dogs and Cats"

_animals, 2022, doi:10.3390/ani12030247_

Round 1

Reviewer 1 Report

Congratulations on this meta-analysis. Even though the number of studies represents a limitation (which you duly mentioned), it is made sufficiently clear that

1) the single points that influence GOR are to be looked at separately

2) clear guidelines are impossible to be formulated

3) what evidence is there (even if only indicative) is presented clearly

We need more studies like this. One probable reason for the small number of meta analyses in veterinary medicine is the low total number and high variability between studies and most topics.

There is one little error: in Table 1 under the study of Viskjer 2017, the average age of the dogs is wrong.

best regards

Author Response

Dear Reviewer,

Thank you so much for your effort and time spent to evaluate our manuscript.

Regarding your comment on the correction of our table: Obviously, the age of the dogs is in months and not in years! Thank you for spotting the error.

Reviewer 2 Report

Dear Authors, this is a useful and interesting contribution to the literature.

I think that as written it is in a good state, with only some minor english revisions required.

I wonder if fig3 could be improved in some way? the subsequent forest plots with the included 'traffic light' data for each study are much more helpful to the reader - perhaps you could dispense with fig 3, or replace it with a smaller figure that only includes studies not included in forest plots?

Author Response

Dear Reviewer,

Thank you very much for your comments.

Regarding Figure 3, it is true that information in this figure are included in the subsequent forest plots. However, we believe that it would be helpful to the reader to see the bias table for the included studies altogether. We could omit this figure if the Editor agrees.